# Measuring organisational readiness for health system change in South Africa using the European Foundation for Quality Management Excellence Model (EFQM)

**Magome Albanos Masike**[1]*, **Ozayr Haroon Mahomed**[2]

**1** Department of Public Health and Nursing, University of Kwazulu-Natal, Durban, KwaZulu-Natal Province, South Africa, **2** Department of Public Health and Nursing, University of Kwazulu-Natal, Durban, KwaZulu-Natal Province, South Africa

* magome61@gmail.com

## Abstract

Successful health system transformation depends on organisational readiness, yet few African studies have applied structured excellence frameworks to assess change capacity. This study evaluated the readiness of two district health systems in South Africa's North West Province using the European Foundation for Quality Management Excellence Model (EFQM). A descriptive cross-sectional survey was conducted using a standardised, EFQM-based, self-administered questionnaire. The questionnaire was distributed to 200 senior and middle-level health managers, chief executive officers, and clinical leaders across the Bojanala Platinum and Ngaka Modiri Molema districts. Quantitative EFQM domain scores were calculated and compared across districts. A total of 193 responses were received (96.5%), including complete European Foundation for Quality Management Excellence Model process data from 112 participants. The combined sample achieved a mean EFQM -PER score of 75.45 (SD = 16.61) and a median of 76.00 (IQR: 66.50–87.25). Bojanala recorded higher EFQM-PER performance (mean = 77.74, SD = 14.88; median = 80.45, IQR: 72.00–89.00) than Ngaka Modiri Molema (mean = 74.06, SD = 17.54; median = 74.80, IQR: 64.00–86.50). Leadership (76.94%) and Policy and Strategy (72.79%) were the strongest-performing EFQM Excellence Model domains, falling within the strengthening range (greater than 70%). People (51.81%) and Partnerships and Resources (45.5%) scored below the readiness threshold, identifying critical vulnerabilities in human resource engagement and resource coordination. No significant associations were observed between readiness scores and socio-demographic characteristics. Health system readiness in the North West province is supported by strong leadership and policy alignment; however, weaknesses in workforce empowerment and partnership management pose substantial risks to sustainable reform. Targeted

**Data availability statement:** All relevant data are within the paper.

**Funding:** The authors received no specific funding for this work.

**Competing interests:** The authors have declared that no competing interests exist.

strengthening of these underperforming domains is essential to translate strategic intent into resilient, people-centred health system change.

## Introduction

Since the advent of democracy in 1994, South Africa has implemented a series of public sector reforms aimed at improving service delivery, including in the health sector. The White Paper on the Transformation of Public Service by the Department of Public Service Administration (DPSA) in 1995 [1] was intended to transform how services are delivered to the population of RSA and provided the overarching framework for sector-specific initiatives.

The White Paper on the transformation of health services provided the framework for delivering health services, based on the District Health System model. Provincial health departments are primarily responsible for providing most public health services, which comprise the bulk of the national health budget [2]. The implementation of this policy framework has been devolved to provincial health departments, each adapting the District Health System to its local population's needs and fiscal capacity.

Despite the presence of a substantial and skilled healthcare workforce, supported by sound policies, several challenges persist in the South African public healthcare system, including budget allocation capacity, inadequate distribution of personnel, the quality of care, low morale among the health workforce, and gaps in leadership and innovation [3].

At the facility level, long waiting times, lower quality of care compared with private healthcare, ageing facilities, and inadequate infection prevention measures impede the attainment of quality-of-care standards [4].

These operational shortcomings are compounded by broader leadership and governance weaknesses across the health system, including limited accountability, inconsistent use of data in decision-making, and inadequate performance management. Collectively, these factors constrain the health system's ability to deliver equitable and efficient care [5–7].

To address these challenges, the government has implemented various programmes and policy adjustments to enhance the effectiveness, safety, and quality of healthcare delivery [8], including the National Health Insurance [9], the introduction of Community Health Workers [10], the rollout of Mobile Clinics [11], the rollout of Immunisation programmes [12], the rollout of Mental Health programmes [13], as well as School Health programmes [14], the Presidential Health Summit [15], and the development of a policy on quality in healthcare for RSA. However, translating these national initiatives into effective district-level implementation remains inconsistent. District management teams play a pivotal role in this process, tasked with adapting national policies into district-specific strategies, operational plans, and budgets [16]. Weak leadership capacity, insufficient managerial support, and unsupportive organisational environments continue to limit their ability to deliver [17].

The persistent difficulties in implementation indicate that leadership and governance are central to a health system's performance [18]. To translate policy goals into measurable improvements, these areas must be strengthened.

Effective leadership involves balancing management capacity across all levels of the health system [19] by (a) ensuring an adequate number of managers at all levels, (b) ensuring that managers have the required competencies, (c) ensuring a support system for all management levels, and (d) ensuring a conducive environment in which managers can excel [20,21]. Healthcare leadership requires adaptive, multi-level approaches to meet the complex demands of transformation [22].

Established leadership models provide a roadmap for managing change. The EFQM Excellence Model serves as a practical tool that builds on these ideas. It offers a comprehensive way to assess an organisation's readiness for change by simultaneously examining leadership, stakeholder involvement, internal processes, and results [23].

The European Foundation for Quality Management (EFQM) Excellence Model is a comprehensive, non-prescriptive management framework, originally developed to promote sustainable organisational excellence and competitiveness through structured self-assessment and continuous improvement.

Its core purpose is to enable any organisation, regardless of sector or size, to manage change effectively, align strategy with execution, and achieve outstanding results by focusing on stakeholder needs and long-term value creation.

Although the EFQM Model was not originally designed for the health sector, it has proven highly applicable as a framework for assessing health systems. This is due to its holistic, systems-thinking orientation, which integrates leadership, process excellence, and multi-stakeholder outcomes, elements that mirror the complex, interdependent nature of healthcare delivery. Empirical applications across European health systems have shown that the model supports the identification of improvement opportunities, fosters a culture of excellence, and correlates with measurable gains in patient satisfaction, reduced waiting times, and organisational performance. Its generic design and RADAR (Results, Approach, Deployment, Assessment and Refinement) logic make it particularly suitable for public-sector health organisations facing resource constraints, regulatory demands, and the need for continuous quality improvement [24,25].

When applied in healthcare leadership, EFQM enables organisations to benchmark current capabilities, identify gaps across the enabler and results domains, and structure action plans for sustainable improvement, making it particularly valuable during large-scale changes such as South Africa's NHI rollout. While EFQM has been widely applied across sectors, its use in assessing South African health districts remains limited.

The North West Province has approximately 4.15 million people, representing 6.7% of the RSA's total population. It spent an estimated R12,274 billion on delivering healthcare services through a network of hospitals, clinics, and community-based services intended to meet the needs of its population [26].

This study aims to conduct a baseline assessment of the health system's readiness to change in the two districts of the North West Province's Department of Health, using the EFQM Excellence Model. The North West Province is a pertinent case for such an assessment, given its predominantly rural population, persistent service delivery inequities, and leadership and governance challenges that mirror those observed nationally. The findings will provide insights into current performance, highlight areas for improvement, and inform the design of context-specific strategies to address existing systemic challenges.

Despite ongoing policy changes and funding, the quality of healthcare still varies widely across South Africa. This suggests that deeper, underlying issues in management and structure are hindering the effective delivery of services. In the North West Province, these challenges are particularly evident in the uneven implementation of national initiatives, limited managerial capacity at the district level, and inconsistent leadership effectiveness, with the rollout of national programmes uneven, management at the local district level often stretched thin, and leadership inconsistent.

As a result, the province is an ideal place to assess how ready a health system is for major change. This is particularly relevant as the country moves forward with the National Health Insurance (NHI). Understanding this context is crucial for

identifying key areas for improvement. The goal is to strengthen local leadership, governance, and management systems to ultimately provide fair and efficient healthcare.

## Methods

### Study design and setting

A descriptive cross-sectional survey was conducted from August to November 2019 in two districts of the North West Province, South Africa: Ngaka Modiri Molema District Municipality (NMMDM) and Bojanala Platinum District Municipality (BPDM). Both districts include a mix of rural and urban areas.

### Study population and sample

The study population comprised senior health professionals directly involved in leadership, governance, and operational management in the two districts. This included facility managers, clinical and programme managers, and district or provincial health officials at the director and chief director levels. Administrative personnel and staff not engaged in management or decision-making were excluded. A census approach was used to ensure comprehensive representation of all eligible personnel across the selected district offices and facilities. Of the 200 questionnaires distributed, 193 were returned, yielding a response rate of 96.5%, facilitated by in-person distribution and collection by the research team.

### Data collection (measures)

Data were collected using a self-administered standard questionnaire. The instrument assessed five core domains of the EFQM Excellence Model.

a. Leadership

b. Strategy

c. People

d. Partnerships and resources

e. Processes, Products and Services

The questionnaire design followed the Likert Scale format. Items were rated on a five-point scale, with options allowing respondents to indicate whether they "strongly agree," "agree," "neutral," "disagree," or "strongly disagree" with the survey statements. Each domain contained multiple statements aligned with EFQM criteria. The instrument was piloted in the Dr Kenneth Kaunda District Municipality to assess clarity, feasibility, and internal consistency before deployment at the study sites. Based on pilot feedback, minor adjustments were made to improve clarity of wording and domain alignment [27].

Questionnaires were delivered and collected in person by the research team to ensure high response rates. All participants received written information sheets outlining the study's purpose, procedures, and their rights, and signed informed consent forms before participation.

### Data management and analysis

Completed questionnaires were coded and entered into Microsoft Excel, then exported to IBM SPSS Statistics version 18.5 for analysis. Data cleaning and verification were conducted to ensure accuracy. Descriptive statistics (frequencies, percentages, means, and standard deviations) were computed for respondent characteristics and EFQM domain scores.

Responses marked "agree" or "strongly agree" were scored 1, and all others 0. Mean scores for each domain were calculated and weighted using EFQM's standard scoring system, which comprises enablers (leadership, strategy, people,

partnerships/resources, and processes) and results (customer results, people results, society results, and key performance results). The overall EFQM score was derived by summing the weighted domain scores.

Descriptive statistics, including frequencies and percentages, means and standard deviations (SD), were used to summarise participant characteristics, particularly when distributions were skewed. In such cases, medians and interquartile ranges (IQR**)** were also employed.

Participants were categorised as having complete or incomplete EFQM data to assess potential selection bias. Sociodemographic characteristics were compared between these two groups using Pearson's chi-square test (or Fisher's exact test where cell counts were small). There were no statistically significant sociodemographic differences between those with complete and incomplete EFQM data; therefore, subsequent EFQM analyses were restricted to those with complete EFQM scores.

### EFQM performance score and sub-criterion comparisons between districts

Normality of the EFQM sco**r**e distribution within each district was assessed using the Shapiro–Wilk test. EFQM scores for BPDM showed a significant deviation from normality, with negative skewness, whereas scores for NMMDM approximated a normal distribution. To ensure robust inference, both parametric and non-parametric tests were used.

For the total EFQM score and each of the five EFQM sub-criteria, we reported both the mean (SD) and the median (IQR) by district and for the combined sample. Differences in mean scores between districts were tested using Welch's independent-samples t-test (which does not assume equal variances). Differences in medians and distributions were examined using the Mann–Whitney U test. District-level distributions of EFQM PER were visualised using box plots and violin–box hybrid plots.

### Dichotomisation of EFQM outcomes and association with sociodemographic factors

To facilitate interpretation, EFQM was dichotomised at the sample median (78.92) to define "high EFQM performance" (≥78.92**)** and **"**lower EFQM performance" (<78.92).

Bivariate associations between the binary EFQM outcome and sociodemographic variables were examined using cross-tabulations, chi-square tests, and calculation of crude prevalence ratios (PRs) with 95% confidence intervals (CIs).

Because high EFQM performance was common in the sample (50%), we used prevalence ratios rather than odds ratios to avoid overestimating associations. Adjusted PRs were estimated using modified Poisson regression with robust standard errors, with the dichotomised EFQM outcome as the dependent variable and gender, age group, race, educational level, salary level/position and district as independent variables. Results are presented as crude and adjusted PRs with 95% CIs and p-values. All tests were two-sided, and $p < 0.05$ was considered statistically significant.

### Reliability

The questionnaire was piloted in the Dr KK Kaunda District Municipality. Cronbach's alpha was used to estimate the reliability of each scale. A Cronbach's alpha value between 0.7 and 0.9 indicates internal consistency. The results show that all Cronbach's alpha values are greater than 0.7, indicating acceptable to excellent internal consistency (≥0.7).

### Ethics

Ethical approval for the study was obtained from the Biomedical Research Ethics Committee of the University of KwaZulu-Natal (BREC reference number BREC/690/18). Gatekeeper permission was granted by the North West Provincial Department of Health. Written informed consent was obtained from all participants before data collection. Participation was voluntary, and respondents could withdraw at any time without penalty. No identifying information was collected, and all

data were stored securely and analysed anonymously. The study was conducted in accordance with the principles of the Declaration of Helsinki [28].

## Results

### Participants characteristics

Demographic data were available for 193 (96%). Complete data for the EFQM process assessment were available for 112 (56%) of the participants.

   Table 1 summarises the socio-demographic profile of participants by EFQM completion status. There were no statistically significant socio-demographic differences between those who completed the EFQM tool in full and those who did not (Table 1). Most respondents (59%) were from Ngaka Modiri Molema District. Most were Black (93%), and 67% of participants were female. Over half (56%, n = 105) held positions at lower management levels (levels 9–10), and 62% (n = 70) had a bachelor's degree or higher.

### Total EFQM scores

Normality of EFQM-PER scores was assessed using skewness and kurtosis, and the Shapiro–Wilk test. There was a statistically significant deviation from normality in the overall sample ($\chi^2(2) = 12.03$, $p = 0.0024$). EFQM scores in Bojanala did not significantly deviate from normality (Shapiro–Wilk W = 0.959, $p = 0.135$), whereas Ngaka Modiri Molema showed a statistically significant departure from normality (Shapiro–Wilk W = 0.951, $p = 0.009$).

   A comparative analysis was conducted using both parametric (Welch's *t-test*) and nonparametric (Mann–Whitney U test) approaches to ensure robust inference, as the normality assumption was violated.

### Overall EFQM performance

The combined sample had a mean EFQM- score of 75.45 (SD = 16.61) and a median of 76.00 (IQR: 66.50–87.25). Bojanala recorded higher EFQM- performance (mean = 77.74, SD = 14.88; median = 80.45, IQR: 72.00–89.00) than Ngaka

**Table 1. Frequency table of Sociodemographic profile of the participants based on completed EFQM survey.**

|  |  | Incomplete EFQM | complete EFQM | Total | Chi squared | p value |
|---|---|---|---|---|---|---|
| **Gender** | Female | 52 (27%) | 76 (40%) | 128 (67%) | 0,17 | 0,68 |
|  | Male | 27 (15%) | 36 (19%) | 64 (33%) |  |  |
| **Age Group** | 25-35 years | 8 (4%) | 16 (8%) | 24 (13%) | 1,52 | 0,68 |
|  | 36-45 years | 21 (11%) | 29 (15%) | 50 (26%) |  |  |
|  | 46-55 years | 32 (17%) | 48 (25%) | 80 (42%) |  |  |
|  | 56 years and above | 18 (9%) | 19 (10%) | 37 (19%) |  |  |
| **Race** | Black African | 77 (40%) | 101 (53%) | 178 (93%) | 2,55 | 0,11 |
|  | Whites/Coloured/Asian | 3 (2%) | 11 (6%) | 14 (7%) |  |  |
| **Salary Level/Position** | Level 9–10 | 44 (23%) | 61 (32%) | 105 (56%) | 0,36 | 0,84 |
|  | Level 11–12 | 21 (11%) | 35 (19%) | 56 (30%) |  |  |
|  | Level 13–14 | 12 (6%) | 16 (8%) | 28 (15%) |  |  |
| **Educational Level** | Senior Certificate | 6 (3%) | 4 (2%) | 10 (5%) | 3,04 | 0,39 |
|  | National Diploma | 30 (16%) | 34 (18%) | 64 (33%) |  |  |
|  | Bachelor's degree | 23 (12%) | 38 (20%) | 61 (32%) |  |  |
|  | Postgraduate | 21 (11%) | 36 (19%) | 57 (30%) |  |  |
| **District** | Bojanala | 36 (19%) | 43 (23%) | 79 (41%) | 0,98 | 0,32 |
|  | Ngaka Modiri Molema | 43 (23%) | 69 (36%) | 112 (59%) |  |  |

Modiri Molema (District 1) (mean = 74.06, SD = 17.54; median = 74.80, IQR: 64.00–86.50). Welch's t-test indicated that the difference in mean EFQM- scores between districts approached statistical significance (t = 1.68, p = 0.096), while the Mann–Whitney U test also suggested a borderline difference that did not reach conventional significance (U = 1244, p = 0.118). The effect size (Cohen's d = 0.22) indicates a small advantage for Bojanala.

Table 2 presents the self-assessment scores across the EFQM variables. Overall, the results suggests that the NDW-DoH is relatively strong in leadership, policy and strategy, and processes, there seems to be challenges in people management and partnership and resource management capabilities..

### EFQM sub-criteria scores

Bojanala Platinum District achieved higher EFQM sub-criteria scores across all domains than Ngaka Modiri Molema District. For Leadership, Bojanala recorded higher mean and median scores than Ngaka Modiri Molema (means 20.65 vs. 19.47; medians 22.0 vs. 20.0). However, this difference was not statistically significant in either Welch's t-test (t ≈ 1.57, p = 0.118) or the Mann–Whitney U test (p = 0.072).

Bojanala achieved statistically significantly higher mean and median scores for both the People domain (mean 14.40 vs 12.67; median 15.20 vs 12.60) and the Strategy domain (mean 12.52 vs 10.45; median 13.50 vs 10.80), with p < 0.01 on both Welch's t-tests and Mann–Whitney U tests.

For Partnerships and Resources and Processes, Products & Services, Bojanala also scored higher on average (Partnerships: 11.82 vs. 10.27; Processes: 23.35 vs. 21.66). However, the strength of the statistical evidence differed by test: mean differences were not significant in Welch's t-tests (p = 0.058 and p = 0.141, respectively), whereas the Mann–Whitney U tests indicated significant group differences for both sub-criteria (p = 0.031 and p = 0.049, respectively).

Table 3 is used for comparative analysis of the two districts are subjects of this research using the EFQM framework and scored across all the sub-criteria.

The BPDM is faring much better than NMMDM across all criteria. What is clear from the scores is that neither of the district municipalities is that they are nowhere near being viewed as excellence.

### EFQM component scores

The self-assessment scores from the North West Province Department of Health delivered a 76.79% for Leadership and a 72.79% for Policy and Strategy. These scores suggests that the department has made telling progress in building key foundational capabilities. This also shows the existence of coherence and intersection needed to drive and sustain transformational change. Leadership as well as policy and strategy intersect mainly in the area "policy leadership." [29]. This is where leaders actively shape, align, and adapt policy environments to the purpose of the organization.

However, it must be remembered that scores are from a self-assessment. They must be accepted with a level of circumspection as these scores may carry inherent bias risks. Some third level validation may be required before they are accepted as is.

The score in the People domain (51.81%) reflects challenges in the area of staff engagement, recognition systems, and capacity-building practices. Although human resource systems are functional, the domain did not perform as highly as Leadership, indicating an opportunity to improve the staff experience, well-being programmes, performance management, and workforce planning.

People (51.81%) and Partnerships & Resources (45.50%) were the two lowest-scoring domains. These results indicate that for the NWDoH to drive true change that will be sustainable, its people and its foundational core must be able to handle and carry the change through. This mirrors a commonly stated challenge in organisational change management where the conditions that are necessary to make change take root often are neglected or not given the correct attention. These areas may be key priorities for strengthening the system.

**Table 2. EFQM components responses.**

| RESEARCH STATEMENTS | STRONGLY AGREE | AGREE | NEUTRAL | DISAGREE | STRONGLY DISAGREE |
|---|---|---|---|---|---|
| **LEADERSHIP** **(n = 555)** | | | | | |
| 1.1a) Leadership develops the Vision, Mission and Values and are role models of a Culture of Excellence. | 47 (42%) | 43 (38%) | 6 (5%) | 8 (7%) | 7 (6%) |
| 1.1b) Leaders in the organisation are personally involved in ensuring the organisation's management system is developed, executed and continuously improved. | 41(37%) | 47(42%) | 6(5%) | 12(11%) | 5(4%) |
| 1.1c) Leaders in the organisation are involved with customers, partners and representatives of the community. | 40(36%) | 50(45%) | 7(6%) | 8(7%) | 6(5%) |
| 1.1d) Leaders in the organisation motivate, support and recognise the organisation's people. | 39(35%) | 39(35%) | 7(6%) | 17 (15%) | 9(8%) |
| 1.1e) Leadership ensures that the organisation is flexible and manages change effectively. | 39(35%) | 42(38%) | 7(6%) | 17 (15%) | 6(5%) |
| | 206/555 (37.12%) | 221/555 (39.82%) | 33/555 (5.95%) | 62/555 (11.17%) | 33/555 (5.95%) |
| | **76.94%** | | **17.12%** | **17.12%** | |
| **POLICY AND STRATEGY** **(n = 555)** | | | | | |
| 2.1 a) The policy and Strategy of the organisation are based on the present and future needs and expectations of stakeholders. | 5(18%) | 63(56%) | | 14(13%) | 20(18%) |
| 2.1 b) The Policy and Strategy of the organization are based on information from performance measurement, research, information and creativity-related activities. | 31(28%) | 58(52%) | 7(6%) | 9(8%) | 6(5%) |
| 2.1 c) The organisation's policy and strategy are constantly developed, reviewed, and updated according to community needs. | 31(28%) | 53(47%) | 10(9%) | 10(9%) | 7(6%) |
| 2.1 d) The organisation's policy and strategy are implemented through a framework of key processes. | 24(21%) | 68(61%) | 11(10%) | 5(4%) | 3(3%) |
| 2.1 e) The Policy and Strategy of the organisation are communicated and effectively implemented | 26(23%) | 45(40%) | 11(10%) | 23(21%) | 6(5%) |
| | 117/555 (21.08%) | 287/555 (51.71%) | 48/555 (8.65%) | 61/555 (10.99%) | 42/555 (7.57%) |
| | **72.79%** | | **8.65%** | **18.56%** | |
| **PEOPLE** **(n = 444)** | | | | | |
| 3.1 a) People resources are planned, managed and improved in the department. | 20(18%) | 51(46%) | 11(10%) | 20(18%) | 9(8%) |
| 3.1 b) People's knowledge and competencies are identified, developed, and sustained. | 18(16%) | 39(35%) | 12(11%) | 31(28%) | 11(10%) |
| 3.1 c) People are involved and empowered in the department. | 18(16%) | 39(35%) | 16(14%) | 27(24%) | 11(10%) |

*(Continued)*

| RESEARCH STATEMENTS | STRONGLY AGREE | AGREE | NEUTRAL | DISAGREE | STRONGLY DISAGREE |
|---|---|---|---|---|---|
| 3.1 d)<br>People are rewarded, recognised and cared for | 12(11%) | 33(29%) | 9(8%) | 34(30%) | 23(21%) |
| | 68/444 (15.32%) | 162/444 (36.49%) | 48/444 (10.81%) | 112/444 (25.23%) | 54/444 (12.16%) |
| | **51.81%** | | **10.81%** | **37.39%** | |
| **PARTNERSHIP AND RESOURCES**<br>**(n = 444)** | | | | | |
| 4.1 a)<br>Finances and SCM processes are appropriately managed. | 7(6%) | 27(24%) | 19(17%) | 40(38%) | 18(16%) |
| 4.1 b)<br>The department always attempts to improve technological Innovation to improve organisational operations. | 10(9%) | 45(40%) | 17(15%) | 29(26%) | 10(9%) |
| 4.1 c)<br>The department consistently implements change management systems to enhance organisational performance. | 9(8%) | 41(37%) | 16(14%) | 36(32%) | 9(8%) |
| 4.1 d)<br>Information and knowledge support effective decision-making and build the organisation's capability. | 14(13%) | 49(44%) | 12(11%) | 29(26%) | 7(6%) |
| | 40/444 (9.01%) | 162/444 (36.49%) | 64/444 (14.41%) | 134/444 (30.18%) | 44/444 (9.91%) |
| | **45.50%** | | **14.41%** | **40.09%** | |
| **PROCESSES**<br>**(n = 444)** | | | | | |
| 5.1 a)<br>Processes are systematically designed and managed. The organisation implements its system and programs in an integrated fashion. | 9(8%) | 54(48%) | 17(15%) | 22(20%) | 9(8%) |
| 5.1 b)<br>Processes are enhanced using creativity to fully satisfy and generate increasing value for customers and other stakeholders. | 10(9%) | 56(50%) | 12(11%) | 26(23%) | 7(6%) |
| 5.1 c)<br>Packages of Services are designed and developed based on customer needs and expectations. | 16(14%) | 60(54%) | 14(13%) | 15(13%) | 6(5%) |
| 5.1 d)<br>Community relationships are managed and enhanced through active participation and consultation. | 16(14%) | 60(54%) | 10(9%) | 19(17%) | 6(5%) |
| | 51/444 (11.49%) | 230/444 (51.80%) | 53/444 (11.94%) | 82/444 (18.47%) | 28/444 (6.31%) |
| | **63.29%** | | **11.94%** | **24.78%** | |

The EFQM Total score (62.07%) indicates a moderate level of organisational excellence, with substantial room for improvement, particularly in domains linked to sustainability, strategic alignment, and resource optimisation.

Fig 1 (the radar plot) highlights a performance habits defined by strong leadership and service delivery, fair people management, and relatively unconvincing human resources management and strategic partnerships.

Table 3. Comparative Analysis - organisational maturity and readiness assessment between BPDM and NMMDM using EFQM.

| EFQM Sub criterion | Max Score | BPDM Mean (SD) | NMMDM Mean (SD) | Combined Mean (SD) | BPDM Median (IQR) | NMMDM Median (IQR) | Combined Median (IQR) | Readiness Category (Combined) | Welch t-test p | Mann–Whitney U p |
|---|---|---|---|---|---|---|---|---|---|---|
| Leadership | 25 | 20.65 (3.75) | 19.47 (4.24) | 19.98 (4.10) | 22 (19 –23) | 20 (17 –22) | 21 (18 –22) | Developing | 0.118 | 0.072 |
| People | 18 | 14.40 (2.81) | 12.67 (3.31) | 13.33 (3.27) | 15.2 (13.6–16.0) | 12.6 (10.8–14.4) | 13.6 (11.2–14.4) | Developing | **0.004** | **0.003** |
| Strategy | 18 | 12.52 (3.21) | 10.45 (3.37) | 11.23 (3.43) | 13.5 (11.7–14.4) | 10.8 (8.1–12.6) | 11.7 (9.0–13.5) | Developing | **0.006** | **0.004** |
| Partnerships & Resources | 18 | 11.82 (3.62) | 10.27 (3.65) | 10.85 (3.70) | 12.6 (10.8–14.4) | 10.8 (8.1–12.6) | 11.7 (9.0–13.5) | Developing | 0.058 | **0.031** |
| Processes, Products & Services | 28 | 23.35 (5.92) | 21.66 (6.41) | 22.32 (6.28) | 25.2 (22.4–28.0) | 22.4 (18.2–25.2) | 23.8 (21.0–26.6) | Developing | 0.141 | **0.049** |
| EFQM–PER Total | **109** | **77.74 (14.88)** | **74.06 (17.54)** | **75.45 (16.61)** | **80.45 (72.05–89.0)** | **74.80 (64.90–86.50)** | **76.0 (69.60–87.30)** | Developing | 0.124 | 0.091 |

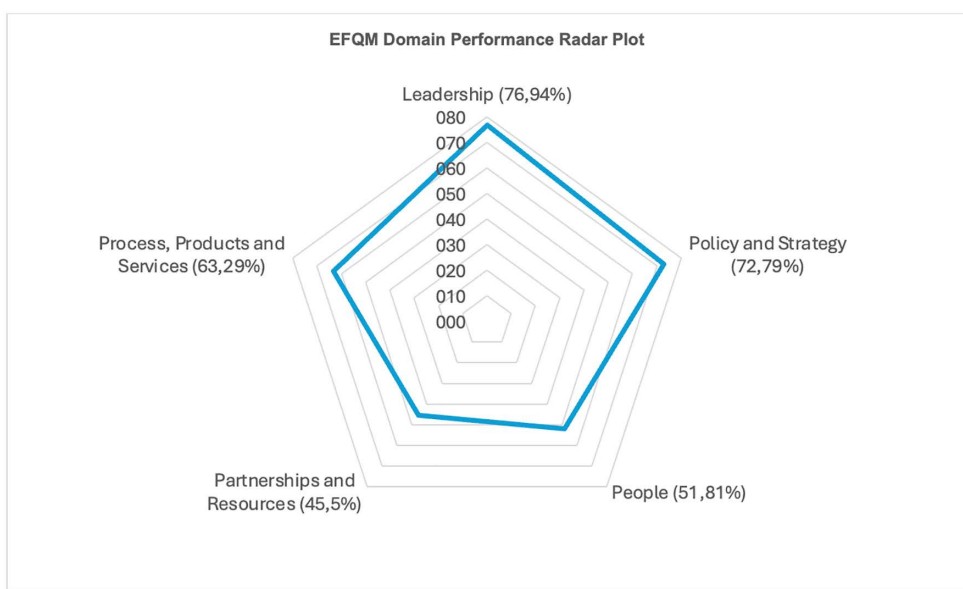

**Fig 1. EFQM domain performance radar plot.**

## Overall EFQM process score

### Association between managers' sociodemographic characteristics and EFQM process scores

A total of 111 healthcare managers were included in the EFQM readiness analysis. Overall, 56 (50.50%) participants scored at or above the median EFQM threshold of 76, while 55 (49.50%) scored below 76.

Table 4 summarises the distribution of EFQM performance across key sociodemographic characteristics. Among female managers, 39 (35%) scored below the median and 36 (32%) above it, compared with 15 (14%) and 21 (19%), respectively, among male managers. Higher EFQM performance was most common among managers aged 46–55 years

**Table 4. Frequency table, Bivariate and Multivariate associations between demographic variables and EFQM performance above the median.**

| Variables | Sub category | EFQM <76 | EFQM >=76 | Total | Chi squared | p value | Prevalence ratio (95% CI) | p value | Adjusted Prevalence ratio | p value |
|---|---|---|---|---|---|---|---|---|---|---|
| Gender | Female | 39 (35%) | 36 (32%) | 75 (68%) | 1,04 | 0,31 | 1 | | | |
| | Male | 15 (14%) | 21 (19%) | 36 (32%) | | | 1,22 (0,84-1,75 | 0,12 | 1,35 (0,91-1,99) | 0,13 |
| Age Group | 25-35 years | 7 (6%) | 9 (8%) | 16 (14%) | 1,6 | 0,66 | | | | |
| | 36-45 years | 17. (15%) | 12 (11%) | 29 (26%) | | | 0,74 (0.40-1,36) | 0,33 | 0,82 (0,45-1,50) | 0,52 |
| | 46-55 years | 22 (29%) | 26 (23%) | 48 (43%) | | | 0,96 (0,58-1,60) | 0,88 | 1,03 (0,61-1,77) | 0,89 |
| | 56 years and above | 8 (7%) | 10 (9%) | 18 (6%) | | | 0,98 (0,54-1,80) | 0,97 | 1,32 (0,68-2,58) | 0,41 |
| Race | Black African | 49 (44%) | 51 (46%) | 100 (90%) | 0,05 | 0,823 | 0,94 (0,53-1,66) | 0,82 | 0,82 (0,45-1,50) | 0,51 |
| | Whites/Coloured/Asian | 5 (5%) | 6 (5%) | 11 (10%) | | | 1 | | | |
| Salary Level/ Position | Level 9–10 | 24 (22%) | 37 (33%) | 61 (55%) | 7,18 | 0,03 | 1 | | | |
| | Level 11–12 | 23 (21%) | 11 (10%) | 34 (31%) | | | 0,54 (0,32-0,90) | 0,02 | 0,56 (0,32-0,98) | 0,04 |
| | Level 13–14 | 7 (8%) | 9 (8%) | 16 (14%) | | | 0,93 (0,58-1,50) | 0,76 | 0,71 (0,39-1,28) | 0,25 |
| Educational Level | Senior Certificate | 1 (1%) | 3 (3%) | 4 (4%) | 1,67 | 0,64 | 1 | | | |
| | National Diploma | 15 (14%) | 19 (17%) | 34 (31%) | | | 0,75 (0,39-1,42) | 0,37 | 0,75 (0,38-1,46) | 0,39 |
| | Bachelor's degree | 21 (19%) | 16 (14%) | 37 (33%) | | | 0,58 (0,29-1,14) | 0,29 | 0,63 (0,30-1,30) | 0,21 |
| | Post graduate | 17 (15%) | 19 (17%) | 36 (32%) | | | 0,70 (0,36-1,34) | 0,37 | 0,71 (0,35-1,47) | 0,84 |
| District | Bojanala | 18 (16%) | 24 (22%) | 42(38%) | 2,69 | 0,1 | 1 | 0,33 | | |
| | Ngaka Modiri Molewa | 36 (32%) | 33 (30%) | 69 (62%) | | | 0,84 (0,44-0,74) | | 0,82 (0,57-1,19) | 0,3 |

(26%), followed by those aged 36–45 years (11%). The distribution of EFQM performance was similar across racial categories, with 51 (46%) Black African managers and 6 (5%) from other racial groups achieving scores ≥76.

Level 9–10 managers formed the largest group, 61 (55%), with 37 (33%) achieving EFQM ≥76. Those in Level 11–12 showed a lower proportion of high EFQM performance, 11 (10%), and Level 13–14 managers represented 9 (8%) of high-scoring participants. A higher proportion of postgraduate-trained managers achieved an EFQM score of 76 or higher, 19 (17%), compared with those holding bachelor's (16–14%) or diploma qualifications, 19 (17%). Managers in the Ngaka Modiri Molema District showed a slightly higher proportion of high EFQM scores, 33 (30%), compared with those in Bojanala, 24 (22%).

## Bivariate & multivariate associations (modified poisson regression)

Prevalence ratios (PRs) were estimated using modified Poisson regression with robust standard errors to assess associations between managers' characteristics and the likelihood of achieving EFQM performance at or above the median (≥76). Table 4 presents unadjusted and adjusted PRs with 95% confidence intervals (CI).

In the unadjusted analysis, female managers had a 22% higher prevalence of high EFQM scores than male managers (aPR = 1.22; 95% CI: 0.84–1.75; p = 0.12). However, the association was not statistically significant and remained non-significant after adjustment (aPR = 1.35; 95% CI: 0.91–1.99; p = 0.13). None of the age groups showed statistically significant associations. The 46–55-year group had a similar prevalence of high EFQM scores to younger groups (aPR = 1.03; 95% CI: 0.61–1.77; p = 0.89). No meaningful association was observed between race and EFQM performance (aPR = 0.82; 95% CI: 0.45–1.50; p = 0.51). Managers at Levels 11–12 were significantly less likely to achieve higher EFQM performance than those at Levels 9–10 in both unadjusted (aPR = 0.54; 95% CI: 0.32–0.90; p = 0.02) and adjusted models (aPR = 0.56; 95% CI: 0.32–0.98; p = 0.04).

High EFQM performance was slightly more prevalent in Ngaka Modiri Molema (aPR = 0.82; 95% CI: 0.57–1.19; p = 0.30), but the association was not statistically significant.

## Discussion

Organisational readiness across the two districts in Northwest Province reflects a pattern typical of transitional health systems. The combined overall EFQM score of 62.07% places performance in the "developing-to-strengthening" band. Both districts demonstrate excellent performance in the Leadership and Processes domain, intermediate performance in the People domain, and poor performance in the People and Partnerships & Resources domains. The Bojanala Platinum District outperformed the Ngaka Modiri Molema District across all domains.

### Leadership

Leadership was the strongest-performing domain in both districts, with a score of 76.94%, in line with international studies. An EFQM-based performance evaluation of a public hospital in Tabriz, Iran, reported Leadership as the highest-scoring enabler criterion (73.6/100) [29]. An EFQM assessment conducted across multiple hospitals in India reported Leadership scores ranging from 55 to 67 points, which were higher than those in other domains [30].

Several EFQM studies suggest that leadership is the primary enabler of sustained improvement and is linked to measurable performance gains. In addition, strong leadership and performance across both districts are shaped by regulatory compliance, including Office of Health Standards Compliance inspections, National Core Standards, and mandatory governance audits, rather than by more participatory or distributive leadership approaches. This may partly explain why the Leadership domain appears relatively strong even where underlying system challenges remain.

Bojanala has maintained consistent senior management structures and more functional District Command Councils, whereas Ngaka Modiri Molema has experienced recurring managerial turnover and critical human resource vacancies [31]. Although the difference between the two districts is not statistically significant, the trend suggests that organisational stability plays an important role in shaping leadership performance.

### Processes, products & services

The Processes, Products & Services domain was the third strongest, scoring 63.29%. The mean process scores for Bojanala were higher than those for Ngaka Modiri Molema (23.35% vs. 21.66%), though the overall picture is mixed. Both districts have a high level of process formalisation. Bojanala has fewer low-performing outliers and more consistently implemented service processes, whereas Ngaka Modiri Molema shows greater operational variability across its facilities.

The high performance in the current study aligns with an EFQM case study from a tertiary hospital pharmacy department in Madrid, which implemented process redesign, protocol standardisation, continuous monitoring, and strengthened clinical governance. These measures improved pharmacotherapy safety, efficiency, and productivity, thereby enhancing EFQM process scores [32].

Similarly, findings from the Basque Health Service showed that organisations that used process-mapping tools, clinical pathways, and standard operating procedures to stabilise care processes across hospitals and primary care had higher enabler scores, including Processes [33]. In contrast to the high performance observed in our study and other European studies, a study at an Iranian emergency department reported a process score of 38.4% [34], whilst a study in a public hospital in Tabriz reported a process score of 59.8/140, about 42.6% of the ideal [35]. These poor scores were attributed to undefined workflows, inadequate coordination mechanisms, and inconsistent protocol application.

The divergence between our findings and the Iranian experience is a direct consequence of South Africa's sustained, mandatory investment in operational compliance. The Ideal Clinic Realisation and Maintenance (ICRM) programme and the enforcement mechanisms of the Office of Health Standards Compliance (OHSC) require facility accreditation by mandating the documentation of Standard Operating Procedures (SOPs).

This creates a structural environment in which process documentation is a necessary precondition, thereby improving performance. Bojanala's superior performance compared with Ngaka Modiri Molema aligns with external assessments that describe more integrated referral pathways, better-supported primary health care (PHC) to hospital interfaces, and more stable facility management in that district, whereas Ngaka Modiri Molema is repeatedly flagged for fragmented services and uneven implementation of standardised care processes [36].

## People

Scores for the People component were similar in Bojanala and Ngaka Modiri Molema (22.88% and 21.38%), with no statistically significant difference between the districts. The People criterion score of 51.81% was very low compared with those reported from Europe, where healthcare organisations report People scores above 80%. In Portugal and Spain, stable staffing, structured career pathways, and mandated professional development systems, along with investment in training and recognition, enhance staff satisfaction and service delivery [37]. In university hospitals in Kerman, People and Partnerships & Resources were the weakest enablers, scoring only about 45.50% [38]. The main reasons cited for this low performance included inadequate training budgets, unclear career paths, and ongoing staff turnover.

A similar pattern was observed at Motahari Hospital in Jahrom, where the people component was the lowest-scoring EFQM domain and showed only modest gains even after targeted projects [39]. Similarly, in an Iranian emergency department, low scores in the people domain were attributed to fragile workforce systems and weak professional development structures [40].

In the Northwest Province, chronic understaffing, high burnout, and high vacancy rates contribute to suboptimal performance. Rural facilities in Ngaka Modiri Molema often operate with clinical vacancy rates exceeding 30%, while urban staff continue to leave for Gauteng or coastal provinces, which offer better infrastructure, schools, and career opportunities. However, a People score of 51.81% reflects two national policies, compulsory community service and the Occupational-Specific Dispensation, as well as the personal commitment of health workers who keep services running despite the pressure.

## Strategy

The Strategy domain ranked second highest, with a mean score of 72.79%. In contrast to our study's findings, several high-income countries report higher performance in this domain. Strategy was among the highest-performing enablers in an Italian acute hospital, driven by sustained investment in structured planning and alignment processes [41,42]. An EFQM-based evaluation of a public hospital in Tabriz, Iran, reported a similar profile, with Strategy scoring 46.8/80 [43]. This finding is not an anomaly but a recurrent theme in EFQM evaluations of complex, bureaucratic health systems.

Bojanala performed significantly better than Ngaka Modiri Molema (12.52 vs 10.45; $p < 0.01$). Bojanala's stronger performance reflects more stable management structures and better integration of district planning with operational realities and the stabilising influence of the tertiary hospital complex, which necessitates more sophisticated planning structures.

The near correlation between a high Leadership score (76.94%) and a slightly lower Strategy score (72.79%) indicates a structural paradox. In mature EFQM implementations, these two domains typically track closely. High-performing organisations achieve strategic alignment by actively involving frontline staff in planning, thereby closing the gap between intent and execution [44]. The North West findings, however, mirror patterns observed in highly bureaucratic, centralised systems.

The public health sector in South Africa is plagued by a "compliance-over-strategy" mindset. Provincial facilities are required to adhere to guidelines set out in the National Department of Health's Annual Performance Plan (APP), which does not allow for any deviation. Strategy becomes an administrative exercise in reporting against predefined targets rather than a dynamic process of local problem-solving [45,46].

Strategy is constrained by a rigid provincial fiscal framework in which district managers have limited discretion over budgets, staffing, or procurement. Many strategic objectives, for example, reducing mortality through emergency care reconfiguration, are simply unattainable in districts where ambulance coverage, clinical staffing, and infrastructure remain inadequate. Ngaka Modiri Molema's lower score reflects this fragility, as managers' capacity to adapt provincial plans to local realities is compromised by rural isolation, and frequent leadership turnover erodes continuity.

## Partnership and resources

Partnerships & Resources was the third-weakest area in the EFQM assessment, with a combined mean score of 63.29%. Bojanala performed significantly better than Ngaka Modiri Molema (11.82 vs 10.27 out of 18 points); however, both districts remain well below the levels achieved in health systems with stable supply chains, predictable budgets, and institutionalised collaboration frameworks. Slightly lower scores of approximately 52–54% were reported for Partnerships & Resources in a multi-hospital evaluation in Kerman [47].

The findings in these hospitals were linked to fragmented procurement structures, limited capital investment, and insufficient staff training in resource management [48]. In contrast to our findings and those in Iran, a longitudinal study of hospital pharmacies in Spain demonstrated that, after the introduction of supplier scorecards, structured maintenance systems, and scheduled equipment audits, scores in this domain rose from 42% to 67% [49].

The historical context of the North West Province explains the persistent weakness. The 2018 Section 100(1)(b) national intervention followed years of procurement failures, widespread irregular expenditure, collapsing infrastructure projects, and recurring medicine stockouts. Although oversight mechanisms have been strengthened since then, recent provincial audit outcomes still document delayed procurement, equipment backlogs, and weak contract management. The EFQM score of 62.07% likely reflects an understanding of required policies rather than consistent, reliable implementation.

Bojanala benefits from its proximity to Gauteng and more established relationships with mining-sector partners, which provide occasional support for equipment and maintenance. By contrast, Ngaka Modiri Molema operates within a more rural economy with fewer external actors and substantial logistical barriers. Yet neither district approaches the EFQM maturity thresholds.

The history of governance volatility and institutional corruption has created an environment of mistrust, deterring private-sector engagement and limiting the system's ability to negotiate strategic collaborations that could bridge the funding and infrastructure gap [50]. Consequently, essential service delivery remains largely dependent on unstable public funding.

Strong leadership and well-designed processes are necessary, but they are also insufficient.

They cannot compensate for a broken procurement system, unreliable funding, or the absence of genuine strategic alliances. Until the deep, systemic failures in resource stewardship and partnership governance are effectively addressed, the high scores in Leadership and Processes will remain indicators of potential rather than drivers of sustained excellence.

Bojanala's comparatively higher score likely reflects structural advantages: proximity to Gauteng-based suppliers, better access to technical support, and occasional supplementation from corporate social investment programmes in the mining sector. By contrast, Ngaka Modiri Molema's vast rural geography, poor road infrastructure, and limited logistics capacity introduce delays at every stage of the distribution chain, from depot to clinic.

## Limitations

This study has several limitations that should be considered when interpreting the findings. First, the research was conducted in only two districts within the North West Province, which may limit the generalisability of the results to other districts within the province or to other provinces in South Africa.

Variations in governance structures, resource allocation, and health system performance across districts and provinces mean that the observed EFQM scores may not reflect the readiness levels of the broader national health system. Second, the cross-sectional survey design captures a single point in time and therefore precludes establishing causal relationships between organisational factors and EFQM scores.

Third, although the overall survey response rate was high, only 56% of respondents completed the EFQM process assessment in full. No statistically significant differences in socio-demographic characteristics were observed between complete and incomplete responders, but non-response bias cannot be ruled out, particularly if those who did not complete the tool differ in unmeasured ways, such as engagement levels or satisfaction with organisational processes.

Fourth, the study relied on self-reported perceptions from senior managers. Such data may be influenced by social desirability bias or by individual interpretations of the EFQM criteria, which could lead to overestimating or underestimating readiness in certain domains. Objective performance indicators or independent assessments were not included, which would have strengthened the validity of the findings.

Fifth, studies [51] suggest that implementing Business Excellence Models such as EFQM is not cheap. These studies contend that implementing the EFQM Model in Low- and Middle-Income Countries (LMICs) often poses significant financial and structural challenges. High costs stem from the need for external expertise, international training, technology, and rigorous assessment processes. Research [52] posits that for organisations in low- to middle-income countries (LMICs) to successfully adopt the EFQM Model, they must shift from a 'certification-driven' mindset to one of gradual, long-term organisational transformation that respects local resource limitations.

Finally, while the EFQM Excellence Model is widely used, it is a generic quality management framework and may not fully capture the unique complexities of health system functioning in resource-constrained, decentralised public health settings such as the North West Province. Interpretation of certain EFQM domains may therefore vary between respondents, potentially affecting comparability.

Future research should address these limitations by including a larger, more diverse sample of districts and provinces, using longitudinal designs to assess changes over time, triangulating self-reported scores with objective performance data, and adapting EFQM measures to better reflect the realities of public health systems in low- and middle-income country contexts.

## Conclusion

The EFQM assessment offers a structured view of organisational readiness within the North West Department of Health, revealing a system with established elements of governance and operational discipline, yet one that continues to struggle with foundational capacity constraints.

Overall performance falls within the moderate range, indicating that the province is positioned for improvement but lacks several enabling conditions necessary to sustain large-scale quality reforms. Bojanala Platinum District consistently achieved higher scores—particularly in Leadership, Strategy, and Partnerships & Resources—indicating comparatively greater managerial continuity and operational stability.

Ngaka Modiri Molema, by contrast, showed weaker performance across most domains, reflecting long-standing staffing shortages, infrastructure deficits, and governance instability. These inter-district differences, despite identical policy environments, underscore the importance of local capacity in determining the effectiveness of system-wide initiatives.

People, Partnerships, and Resources emerged as persistent bottlenecks. Workforce shortages, uneven distribution of skills, burnout, and limited professional development continue to undermine organisational resilience. Likewise, weak procurement systems, fragile external partnerships, and inconsistent resource stewardship constrain facilities' ability to implement strategic priorities. Unless these structural issues are addressed, gains in leadership intent or process standardisation will struggle to translate into durable improvements in service delivery.

## Implications for policy

The study's findings point to several policy priorities that should guide future provincial and national interventions. First, workforce strengthening must be treated as a strategic priority rather than an administrative function. This includes targeted recruitment in high-shortage areas, expanded training and career progression opportunities, and structured programmes to reduce burnout and improve staff retention. These interventions are not merely supportive—they are essential to stabilising the People domain and enabling progress across all other EFQM criteria.

Second, investment in supply chain governance, procurement reform, and strategic partnerships is critical. The province's history of financial mismanagement and inconsistent contract oversight has created a fragile resource environment that undermines operational planning. Rebuilding this domain will require transparent procurement systems, clear accountability mechanisms, and long-term partnerships with academic institutions, NGOs, and private-sector entities.

Third, district-level autonomy and adaptive planning capacity need strengthening. The disparity between Bojanala and Ngaka Modiri Molema shows that strategic plans are only as effective as the structures that implement them. Granting districts greater flexibility to tailor provincial priorities to local needs—supported by adequate resources and consistent managerial leadership—could improve strategic alignment and operational execution.

Finally, future research and monitoring systems should incorporate objective performance indicators alongside EFQM self-assessments. Longitudinal designs spanning multiple provinces would help determine whether improvements in the leadership, strategy, and process domains translate into measurable gains in service quality.

Taken together, these policy actions can help bridge the gap between strategic intent and operational delivery, enabling the North West Department of Health to build a more resilient, equitable, and high-performing health system.

## Acknowledgments

The authors express their gratitude to the NWDoH management, Prof. Zuma (HSRC), and the participants for their collaboration and contributions to this work.

## Author contributions

**Conceptualization:** Magome Albanos Masike.

**Data curation:** Magome Albanos Masike.

**Formal analysis:** Magome Albanos Masike.

**Funding acquisition:** Magome Albanos Masike.

**Investigation:** Magome Albanos Masike.

**Methodology:** Magome Albanos Masike.

**Project administration:** Magome Albanos Masike.

**Resources:** Magome Albanos Masike.

**Software:** Magome Albanos Masike.

**Supervision:** Ozayr Haroon Mohamed.

**Validation:** Magome Albanos Masike.

**Visualization:** Magome Albanos Masike.

**Writing – original draft:** Magome Albanos Masike.

**Writing – review & editing:** Magome Albanos Masike.

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
