## [Decision Letter · Decision Letter 0]

22 Feb 2026

PGPH-D-25-03981

Measuring Organisational Readiness for Health System Change in South Africa Using the European Foundation for Quality Management Excellence Model

Dear Dr. Masike,

Thank you for submitting your manuscript to PLOS Global Public Health. After careful consideration, we feel that it has merit but does not fully meet PLOS Global Public Health’s publication criteria as it currently stands. Therefore, we invite you to submit a revised version of the manuscript that addresses the points raised during the review process.

We look forward to receiving your revised manuscript.

Kind regards,

Somayeh Hessam

Academic Editor

Journal Requirements:

1. In the online submission form, you indicated that “All Data Used to arrive at the Findings and Conclusions will be made available at a point of request.”.

3. Uploaded as supplementary information.

Additional Editor Comments (if provided):

Reviewers' comments:

Reviewer's Responses to Questions

**Comments to the Author**

1. Does this manuscript meet PLOS Global Public Health’s publication criteria? Is the manuscript technically sound, and do the data support the conclusions? The manuscript must describe methodologically and ethically rigorous research with conclusions that are appropriately drawn based on the data presented.

Reviewer #1: Yes

Reviewer #2: Partly

Reviewer #3: Yes

2. Has the statistical analysis been performed appropriately and rigorously?

Reviewer #1: Yes

Reviewer #2: Yes

Reviewer #3: Yes

3. Have the authors made all data underlying the findings in their manuscript fully available (please refer to the Data Availability Statement at the start of the manuscript PDF file)?

Reviewer #1: No

Reviewer #2: Yes

Reviewer #3: Yes

4. Is the manuscript presented in an intelligible fashion and written in standard English?

Reviewer #1: Yes

Reviewer #2: Yes

Reviewer #3: Yes

5. Review Comments to the Author

Reviewer #1: The manuscript is sound, well written and presented. From a statistical point of view, statistical analysis was performed appropriately and rigorously for a descriptive study. However, data used for the analysis should be fully made available (not at a point of request as stated in the paper).

The figure titled "EFQM Domain Performance Radar Plot" after line 318 should be numbered e.g. Figure 1. Also Table 3 has been numbered repeatedly in line 292 and line 364.

Reviewer #2: The research targets a highly relevant public health issue by evaluating structural readiness for large-scale changes like South Africa's National Health Insurance (NHI) rollout. The data collection process was highly effective, achieving an initial response rate of 96.5% with 193 returned questionnaires out of 200 distributed. The authors employed a rigorous statistical approach by testing for normality and subsequently using both parametric (Welch's t-test) and non-parametric (Mann-Whitney U) tests to ensure robust inference. The application of a modified Poisson regression with robust standard errors to calculate prevalence ratios is a sound methodological choice, preventing the overestimation of associations common with odds ratios in high-prevalence scenarios.

The are a few things that need some work: The abstract states that Leadership scored 76.9%, Policy and Strategy scored 72.8%, People scored 51.8%, and Partnerships and Resources scored 45.5%. Table 2 aligns with the abstract, listing Leadership at 76.94%, Policy and Strategy at 72.79%, People at 51.71%, Partnerships and Resources at 45.50%, and Processes at 63.29%. However, the narrative under "EFQM Component Scores" completely contradicts these figures, claiming Leadership scored 79.9%, Processes scored 79.7%, People scored 74.1%, Strategy scored 62.4%, and Partnerships & Resources scored 60.3%. The "EFQM Domain Performance Radar Plot" visually and textually supports the incorrect narrative numbers rather than the primary data in Table 2. There is a sequencing error with the tables; the designation "Table 3" is used twice for entirely different datasets (once for organizational maturity comparisons and once for multivariate associations). An article by Oleribe et al. has the wrong reference year (2029, instead of 2019).

Reviewer #3: The manuscript is well-articulated and thoughtfully structured, addressing a significant and under-explored aspect of health system transformation in the South African context. The rationale for applying the EFQM Excellence Model to organisational readiness is clearly developed, and the study demonstrates strong methodological transparency, particularly in its analytic approach and reporting of statistical assumptions. The authors’ decision to use both parametric and non-parametric analyses when normality assumptions were violated enhances confidence in the robustness of the findings.

The discussion section is particularly compelling, as it extends beyond descriptive reporting to provide a nuanced interpretation of system dynamics, including the tension between strong leadership scores and comparatively weaker strategy and resource domains. The identification of a potential “compliance-over-strategy” paradox offers a valuable conceptual contribution that may have relevance beyond the immediate study setting. Furthermore, the limitations are appropriately acknowledged, and the policy implications are clearly articulated in a manner that is actionable for district-level and provincial stakeholders.

As a minor suggestion, the authors could briefly expand on how EFQM domains might be adapted or operationalised differently within resource-constrained public health systems, as this would further enhance the paper’s relevance to other low- and middle-income contexts. Overall, the study is rigorous, accessible, and makes a meaningful contribution to the literature on organizational readiness and health system reform.

6. PLOS authors have the option to publish the peer review history of their article (what does this mean?). If published, this will include your full peer review and any attached files.

**Do you want your identity to be public for this peer review?** For information about this choice, including consent withdrawal, please see our Privacy Policy.

Reviewer #1: No

Reviewer #2: No

Reviewer #3: No

 Figure Resubmissions:

---

## [Decision Letter · Decision Letter 1]

5 Apr 2026

PGPH-D-25-03981R1

Measuring Organisational Readiness for Health System Change in South Africa Using the European Foundation for Quality Management Excellence Model

Dear Dr. Masike,

Thank you for submitting your manuscript to PLOS Global Public Health. After careful consideration, we feel that it has merit but does not fully meet PLOS Global Public Health’s publication criteria as it currently stands. Therefore, we invite you to submit a revised version of the manuscript that addresses the points raised during the review process.

We look forward to receiving your revised manuscript.

Kind regards,

Somayeh Hessam

Academic Editor

**Journal Requirements:**

**Additional Editor Comments (if provided):**

Reviewers' comments:

Reviewer's Responses to Questions

**Comments to the Author**

1. If the authors have adequately addressed your comments raised in a previous round of review and you feel that this manuscript is now acceptable for publication, you may indicate that here to bypass the “Comments to the Author” section, enter your conflict of interest statement in the “Confidential to Editor” section, and submit your "Accept" recommendation.

Reviewer #1: All comments have been addressed

Reviewer #3: All comments have been addressed

Reviewer #4: (No Response)

2. Does this manuscript meet PLOS Global Public Health’s publication criteria? Is the manuscript technically sound, and do the data support the conclusions? The manuscript must describe methodologically and ethically rigorous research with conclusions that are appropriately drawn based on the data presented.

Reviewer #1: Yes

Reviewer #3: Yes

Reviewer #4: Yes

3. Has the statistical analysis been performed appropriately and rigorously?

Reviewer #1: Yes

Reviewer #3: Yes

Reviewer #4: Yes

4. Have the authors made all data underlying the findings in their manuscript fully available (please refer to the Data Availability Statement at the start of the manuscript PDF file)?

Reviewer #1: Yes

Reviewer #3: Yes

Reviewer #4: No

5. Is the manuscript presented in an intelligible fashion and written in standard English?

Reviewer #1: Yes

Reviewer #3: Yes

Reviewer #4: Yes

6. Review Comments to the Author

**Reviewer #1:** All concerns raised in the previous review have been addressed.

**Reviewer #3:** The authors have responded thoughtfully, and the revision shows meaningful improvement. Adding the "Contextual Adaptation of EFQM in Resource-Constrained Public Health Systems" section (Discussion) directly addresses my earlier suggestion, increasing the paper's relevance to other low- and middle-income settings. The argument that low scores may reflect structural constraints rather than poor management is well-articulated and conceptually valuable.

The new paragraph on EFQM implementation costs in LMICs is appropriate. A few minor issues remain before acceptance:

1. Data-narrative mismatch (Partnerships & Resources discussion): The revised text states that Partnerships & Resources had "a combined mean score of 63.29%." This figure matches the Processes domain in Table 2, while Partnerships & Resources is reported as 45.50%. This error appears to persist despite previous corrections and should be addressed, as the interpretation is currently based on the wrong value.

2. Mischaracterization of domain ranking (EFQM Component Scores section): The text states that "Strategy (72.79%) and Partnerships & Resources (45.5%) were the two lowest-scoring domains." However, Strategy is the second-highest domain. The lowest-scoring domains are People (51.81%) and Partnerships & Resources (45.50%). Please correct this.

3. Reference alignment in the new limitation paragraph: The citation to reference (26) on mitigating EFQM implementation costs may not match the referenced source (Tóth & Szűcs, 2025, about EFQM and leadership/innovation). Please verify that the correct source is cited.

4. Minor editorial items: The abbreviation "apar" is used inconsistently in the multivariate results section (sometimes "apar," sometimes "PR" for the same adjusted measure). Decimal notation is also inconsistent (e.g., "51,81%" vs. "45.5%"). Please standardize both throughout.

These issues are easily correctable and do not affect the paper's quality. I recommend acceptance after these editorial corrections and am confident these revisions will further enhance the manuscript's impact and usefulness.

**Reviewer #4:** Data Availability Statement:

The authors state that “all data used to arrive at the findings and conclusions will be made available upon request.” This approach does not align with PLOS guidelines, which require that data be made openly available without restriction. The authors should ensure compliance with the journal’s data sharing policy as applicable.

Background:

The manuscript would benefit from greater clarity regarding the EFQM framework. The authors should spell out EFQM in full at first mention and provide additional context on its purpose and core domains. It would also strengthen the paper to explain why EFQM is an appropriate framework for assessing health systems, particularly given that it was not originally developed for this purpose. Any adaptations made to apply the framework in this context should be clearly described.

Results:

Table 2 would benefit from improved formatting. The current layout, particularly the spacing and section breaks, makes the information difficult to follow. Revising the table for clarity and readability would enhance the presentation of the results.

7. PLOS authors have the option to publish the peer review history of their article (what does this mean?). If published, this will include your full peer review and any attached files.

**Do you want your identity to be public for this peer review?** For information about this choice, including consent withdrawal, please see our Privacy Policy.

Reviewer #1: No

Reviewer #3: No

Reviewer #4: No

**Figure Resubmissions:**

---

## [Decision Letter · Decision Letter 2]

12 May 2026

Measuring Organisational Readiness for Health System Change in South Africa Using the European Foundation for Quality Management Excellence Model

PGPH-D-25-03981R2

Dear Dr Masike,

We are pleased to inform you that your manuscript 'Measuring Organisational Readiness for Health System Change in South Africa Using the European Foundation for Quality Management Excellence Model' has been provisionally accepted for publication in PLOS Global Public Health.

Best regards,

Somayeh Hessam

Academic Editor

Reviewer Comments (if any, and for reference):

Reviewer's Responses to Questions

**Comments to the Author**

1. If the authors have adequately addressed your comments raised in a previous round of review and you feel that this manuscript is now acceptable for publication, you may indicate that here to bypass the “Comments to the Author” section, enter your conflict of interest statement in the “Confidential to Editor” section, and submit your "Accept" recommendation.

Reviewer #1: All comments have been addressed

Reviewer #4: All comments have been addressed

2. Does this manuscript meet PLOS Global Public Health’s publication criteria? Is the manuscript technically sound, and do the data support the conclusions? The manuscript must describe methodologically and ethically rigorous research with conclusions that are appropriately drawn based on the data presented.

Reviewer #1: Yes

Reviewer #4: (No Response)

3. Has the statistical analysis been performed appropriately and rigorously?

Reviewer #1: Yes

Reviewer #4: (No Response)

4. Have the authors made all data underlying the findings in their manuscript fully available (please refer to the Data Availability Statement at the start of the manuscript PDF file)?

Reviewer #1: Yes

Reviewer #4: (No Response)

5. Is the manuscript presented in an intelligible fashion and written in standard English?

Reviewer #1: Yes

Reviewer #4: (No Response)

6. Review Comments to the Author

Reviewer #1: Authors have adequately addressed all comments raised in the previous round of review, hence, the manuscript is acceptable for publication.

Reviewer #4: (No Response)

7. PLOS authors have the option to publish the peer review history of their article (what does this mean?). If published, this will include your full peer review and any attached files.

**Do you want your identity to be public for this peer review?** For information about this choice, including consent withdrawal, please see our Privacy Policy.

Reviewer #1: No

Reviewer #4: No
